# Neural Combinatorial Optimization with Reinforcement Learning

**Irwan Bello**[*]**, Hieu Pham**[*]**, Quoc V. Le, Mohammad Norouzi, Samy Bengio**
Google Brain
{ibello,hyhieu,qvl,mnorouzi,bengio}@google.com

### Abstract

This paper presents a framework to tackle combinatorial optimization problems using neural networks and reinforcement learning. We focus on the traveling salesman problem (TSP) and train a recurrent neural network that, given a set of city coordinates, predicts a distribution over different city permutations. Using negative tour length as the reward signal, we optimize the parameters of the recurrent neural network using a policy gradient method. We compare learning the network parameters on a set of training graphs against learning them on individual test graphs. Without much engineering and heuristic designing, Neural Combinatorial Optimization achieves close to optimal results on 2D Euclidean graphs with up to 100 nodes. Applied to the KnapSack, another NP-hard problem, the same method obtains optimal solutions for instances with up to 200 items. These results, albeit still far from state-of-the-art, give insights into how neural networks can be used as a general tool for tackling combinatorial optimization problems.

## 1 Introduction

*Combinatorial optimization* is a fundamental problem in computer science. A canonical example is the *traveling salesman problem (TSP)*, where given a graph, one needs to search the space of permutations to find an optimal sequence of nodes with minimal total edge weights (tour length). The TSP and its variants have myriad applications in planning, manufacturing, genetics, *etc.* (see (Applegate et al., 2011) for an overview).

Finding the optimal TSP solution is NP-hard, even in the two-dimensional Euclidean case (Papadimitriou, 1977), where the nodes are 2D points and edge weights are Euclidean distances between pairs of points. In practice, TSP solvers rely on handcrafted heuristics that guide their search procedures to find competitive (and in many cases optimal) tours efficiently. Even though these heuristics work well on TSP, once the problem statement changes slightly, they need to be revised. In contrast, machine learning methods have the potential to be applicable across many optimization tasks by automatically discovering their own heuristics based on the training data, thus requiring less hand-engineering than solvers that are optimized for one task only.

While most successful machine learning techniques fall into the family of supervised learning, where a mapping from training inputs to outputs is learned, supervised learning is not applicable to most combinatorial optimization problems because one does not have access to optimal labels. However, one can compare the quality of a set of solutions using a verifier, and provide some reward feedbacks to a learning algorithm. Hence, we follow the reinforcement learning (RL) paradigm to tackle combinatorial optimization. We empirically demonstrate that, even when using optimal solutions as labeled data to optimize a supervised mapping, the generalization is rather poor compared to an RL agent that explores different tours and observes their corresponding rewards.

We propose Neural Combinatorial Optimization, a framework to tackle combinatorial optimization problems using reinforcement learning and neural networks. We consider two approaches based on policy gradients (Williams, 1992). The first approach, called *RL pretraining*, uses a training set to optimize a recurrent neural network (RNN) that parameterizes a stochastic policy over solutions, using the expected reward as objective. At test time, the policy is fixed, and one performs inference

---

[*]Equal contributions. Members of the Google Brain Residency program (g.co/brainresidency).

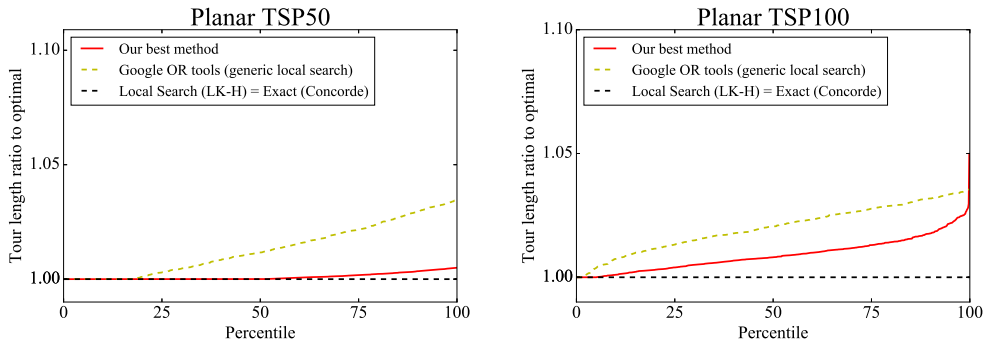

Figure 1: Tour length ratios of LK-H (Helsgaun, 2000) local search and our best method (RL pretraining-Active Search) against optimality, guaranteed by Concorde (Applegate et al., 2006). Generic local search, obtained via Googles vehicle routing problem solver (Google, 2016), applies a set of heuristics starting from the (Christofides, 1976) solution. Note that our method is five orders of magnitude slower than LK-H and Concorde.

by greedy decoding or sampling. The second approach, called *active search*, involves no pretraining. It starts from a random policy and iteratively optimizes the RNN parameters on a single test instance, again using the expected reward objective, while keeping track of the best solution sampled during the search. We find that combining RL pretraining and active search works best in practice.

On 2D Euclidean graphs with up to 100 nodes, Neural Combinatorial Optimization significantly outperforms the supervised learning approach to the TSP (Vinyals et al., 2015b) and obtains close to optimal results when allowed more computation time (see Figure 1). We illustrate the flexibility of the method by also applying it to the KnapSack problem, for which we get optimal results for instances with up to 200 items. Our results, while still inferior to the state-of-the-art in many dimensions (such as speed, scale and performance), give insights into how neural networks can be used as a general tool for tackling combinatorial optimization problems, especially those that are difficult to design heuristics for.

## 2 PREVIOUS WORK

The Traveling Salesman Problem is a well studied combinatorial optimization problem and many exact or approximate algorithms have been proposed for both Euclidean and non-Euclidean graphs. Christofides (1976) proposes a heuristic algorithm that involves computing a minimum-spanning tree and a minimum-weight perfect matching. The algorithm has polynomial running time and returns solutions that are guaranteed to be within a factor of $1.5\times$ to optimality in the metric instance of the TSP.

The best known exact dynamic programming algorithm for TSP has a complexity of $\Theta(2^n n^2)$, making it infeasible to scale up to large instances, say with 40 points. Nevertheless, state of the art TSP solvers, thanks to carefully handcrafted heuristics that describe how to navigate the space of feasible solutions in an efficient manner, can solve symmetric TSP instances with thousands of nodes. Concorde (Applegate et al., 2006), widely accepted as one of the best exact TSP solvers, makes use of cutting plane algorithms (Dantzig et al., 1954; Padberg & Rinaldi, 1990; Applegate et al., 2003), iteratively solving linear programming relaxations of the TSP, in conjunction with a branch-and-bound approach that prunes parts of the search space that provably will not contain an optimal solution. Similarly, the Lin-Kernighan-Helsgaun heuristic (Helsgaun, 2000), inspired from the Lin-Kernighan heuristic (Lin & Kernighan, 1973), is a state of the art approximate search heuristic for the symmetric TSP and has been shown to solve instances with hundreds of nodes to optimality.

More generic solvers, such as Google's vehicle routing problem solver (Google, 2016) that tackles a superset of the TSP, typically rely on a combination of local search algorithms and metaheuristics. Local search algorithms apply a specified set of local move operators on candidate solutions, based

on hand-engineered heuristics such as 2-opt (Johnson, 1990), to navigate from solution to solution in the search space. A metaheuristic is then applied to propose uphill moves and escape local optima. A popular choice of metaheuristic for the TSP and its variants is guided local search (Voudouris & Tsang, 1999), which moves out of a local minimum by penalizing particular solution features that it considers should not occur in a good solution.

The difficulty in applying existing search heuristics to newly encountered problems - or even new instances of a similar problem - is a well-known challenge that stems from the *No Free Lunch theorem* (Wolpert & Macready, 1997). Because all search algorithms have the same performance when averaged over all problems, one must appropriately rely on a prior over problems when selecting a search algorithm to guarantee performance. This challenge has fostered interest in raising the level of generality at which optimization systems operate (Burke et al., 2003) and is the underlying motivation behind hyper-heuristics, defined as "search method[s] or learning mechanism[s] for selecting or generating heuristics to solve computation search problems". Hyper-heuristics aim to be easier to use than problem specific methods by partially abstracting away the knowledge intensive process of selecting heuristics given a combinatorial problem and have been shown to successfully combine human-defined heuristics in superior ways across many tasks (see (Burke et al., 2013) for a survey). However, hyper-heuristics operate on the search space of heuristics, rather than the search space of solutions, therefore still initially relying on human created heuristics.

The application of neural networks to combinatorial optimization has a distinguished history, where the majority of research focuses on the Traveling Salesman Problem (Smith, 1999). One of the earliest proposals is the use of Hopfield networks (Hopfield & Tank, 1985) for the TSP. The authors modify the network's energy function to make it equivalent to TSP objective and use Lagrange multipliers to penalize the violations of the problem's constraints. A limitation of this approach is that it is sensitive to hyperparameters and parameter initialization as analyzed by (Wilson & Pawley, 1988). Overcoming this limitation is central to the subsequent work in the field, especially by (Aiyer et al., 1990; Gee, 1993). Parallel to the development of Hopfield networks is the work on using deformable template models to solve TSP. Perhaps most prominent is the invention of Elastic Nets as a means to solve TSP (Durbin, 1987), and the application of Self Organizing Map to TSP (Fort, 1988; Angeniol et al., 1988; Kohonen, 1990). Addressing the limitations of deformable template models is central to the following work in this area (Burke, 1994; Favata & Walker, 1991; Vakhutinsky & Golden, 1995). Even though these neural networks have many appealing properties, they are still limited as research work. When being carefully benchmarked, they have not yielded satisfying results compared to algorithmic methods (Sarwar & Bhatti, 2012; La Maire & Mladenov, 2012). Perhaps due to the negative results, this research direction is largely overlooked since the turn of the century.

Motivated by the recent advancements in sequence-to-sequence learning (Sutskever et al., 2014), neural networks are again the subject of study for optimization in various domains (Yutian et al., 2016), including discrete ones (Zoph & Le, 2016). In particular, the TSP is revisited in the introduction of Pointer Networks (Vinyals et al., 2015b), where a recurrent network with non-parametric softmaxes is trained in a supervised manner to predict the sequence of visited cities. Despite architecural improvements, their models were trained using supervised signals given by an approximate solver.

## 3 NEURAL NETWORK ARCHITECTURE FOR TSP

We focus on the 2D Euclidean TSP in this paper. Given an input graph, represented as a sequence of $n$ cities in a two dimensional space $s = \{\mathbf{x}_i\}_{i=1}^n$ where each $\mathbf{x}_i \in \mathbb{R}^2$, we are concerned with finding a permutation of the points $\pi$, termed a tour, that visits each city once and has the minimum total length. We define the length of a tour defined by a permutation $\pi$ as

$$L(\pi \mid s) = \left\| \mathbf{x}_{\pi(n)} - \mathbf{x}_{\pi(1)} \right\|_2 + \sum_{i=1}^{n-1} \left\| \mathbf{x}_{\pi(i)} - \mathbf{x}_{\pi(i+1)} \right\|_2, \tag{1}$$

where $\|\cdot\|_2$ denotes $\ell_2$ norm.

We aim to learn the parameters of a stochastic policy $p(\pi \mid s)$ that given an input set of points $s$, assigns high probabilities to short tours and low probabilities to long tours. Our neural network

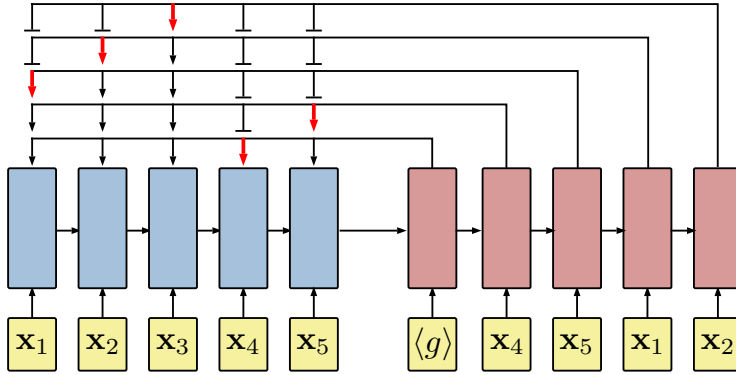

Figure 2: A pointer network architecture introduced by (Vinyals et al., 2015b).

architecture uses the chain rule to factorize the probability of a tour as

$$p(\pi \mid s) = \prod_{i=1}^{n} p\left(\pi(i) \mid \pi(< i), s\right) \ , \tag{2}$$

and then uses individual softmax modules to represent each term on the RHS of (2).

We are inspired by previous work (Sutskever et al., 2014) that makes use of the same factorization based on the chain rule to address sequence to sequence problems like machine translation. One can use a vanilla sequence to sequence model to address the TSP where the output vocabulary is $\{1, 2, \ldots, n\}$. However, there are two major issues with this approach: (1) networks trained in this fashion cannot generalize to inputs with more than $n$ cities. (2) one needs to have access to ground-truth output permutations to optimize the parameters with conditional log-likelihood. We address both isssues in this paper.

For generalization beyond a pre-specified graph size, we follow the approach of (Vinyals et al., 2015b), which makes use of a set of non-parameteric softmax modules, resembling the attention mechanism from (Bahdanau et al., 2015). This approach, named *pointer network*, allows the model to effectively point to a specific position in the input sequence rather than predicting an index value from a fixed-size vocabulary. We employ the pointer network architecture, depicted in Figure 2, as our policy model to parameterize $p(\pi \mid s)$.

### 3.1 ARCHITECTURE DETAILS

Our pointer network comprises two recurrent neural network (RNN) modules, encoder and decoder, both of which consist of Long Short-Term Memory (LSTM) cells (Hochreiter & Schmidhuber, 1997). The encoder network reads the input sequence $s$, one city at a time, and transforms it into a sequence of latent memory states $\{enc_i\}_{i=1}^{n}$ where $enc_i \in \mathbb{R}^d$. The input to the encoder network at time step $i$ is a $d$-dimensional embedding of a 2D point $\mathbf{x}_i$, which is obtained via a linear transformation of $\mathbf{x}_i$ shared across all input steps. The decoder network also maintains its latent memory states $\{dec_i\}_{i=1}^{n}$ where $dec_i \in \mathbb{R}^d$ and, at each step $i$, uses a pointing mechanism to produce a distribution over the next city to visit in the tour. Once the next city is selected, it is passed as the input to the next decoder step. The input of the first decoder step (denoted by $\langle g \rangle$ in Figure 2) is a d-dimensional vector treated as a trainable parameter of our neural network.

Our attention function, formally defined in Appendix A.1, takes as input a query vector $q = dec_i \in \mathbb{R}^d$ and a set of reference vectors $ref = \{enc_1, \ldots, enc_k\}$ where $enc_i \in \mathbb{R}^d$, and predicts a distribution $A(ref, q)$ over the set of $k$ references. This probability distribution represents the degree to which the model is pointing to reference $r_i$ upon seeing query $q$.

Vinyals et al. (2015a) also suggest including some additional computation steps, named *glimpses*, to aggregate the contributions of different parts of the input sequence, very much like (Bahdanau et al., 2015). We discuss this approach in details in Appendix A.1. In our experiments, we find that utilizing one glimpse in the pointing mechanism yields performance gains at an insignificant cost latency.

---

**Algorithm 1** Actor-critic training

---

1: **procedure** TRAIN(training set $S$, number of training steps $T$, batch size $B$)
2: Initialize pointer network params $\theta$
3: Initialize critic network params $\theta_v$
4: **for** $t = 1$ to $T$ **do**
5: $s_i \sim$ SAMPLEINPUT($S$) for $i \in \{1, \dots, B\}$
6: $\pi_i \sim$ SAMPLESOLUTION($p_\theta(.|s_i)$) for $i \in \{1, \dots, B\}$
7: $b_i \leftarrow b_{\theta_v}(s_i)$ for $i \in \{1, \dots, B\}$
8: $g_\theta \leftarrow \frac{1}{B} \sum_{i=1}^{B} (L(\pi_i|s_i) - b_i) \nabla_\theta \log p_\theta(\pi_i|s_i)$
9: $\mathcal{L}_v \leftarrow \frac{1}{B} \sum_{i=1}^{B} \|b_i - L(\pi_i)\|_2^2$
10: $\theta \leftarrow$ ADAM($\theta, g_\theta$)
11: $\theta_v \leftarrow$ ADAM($\theta_v, \nabla_{\theta_v} \mathcal{L}_v$)
12: **end for**
13: **return** $\theta$
14: **end procedure**

---

## 4 OPTIMIZATION WITH POLICY GRADIENTS

Vinyals et al. (2015b) proposes training a pointer network using a supervised loss function comprising conditional log-likelihood, which factors into a cross entropy objective between the network's output probabilities and the targets provided by a TSP solver. Learning from examples in such a way is undesirable for NP-hard problems because (1) the performance of the model is tied to the quality of the supervised labels, (2) getting high-quality labeled data is expensive and may be infeasible for new problem statements, (3) one cares about finding a competitive solution more than replicating the results of another algorithm.

By contrast, we believe Reinforcement Learning (RL) provides an appropriate paradigm for training neural networks for combinatorial optimization, especially because these problems have relatively simple reward mechanisms that could be even used at test time. We hence propose to use model-free policy-based Reinforcement Learning to optimize the parameters of a pointer network denoted $\boldsymbol{\theta}$. Our training objective is the expected tour length which, given an input graph $s$, is defined as

$$J(\boldsymbol{\theta} \mid s) = \mathbb{E}_{\pi \sim p_\theta(.|s)} L(\pi \mid s) . \tag{3}$$

During training, our graphs are drawn from a distribution $\mathcal{S}$, and the total training objective involves sampling from the distribution of graphs, *i.e.* $J(\boldsymbol{\theta}) = \mathbb{E}_{s \sim \mathcal{S}} J(\boldsymbol{\theta} \mid s) .$

We resort to policy gradient methods and stochastic gradient descent to optimize the parameters. The gradient of (3) is formulated using the well-known REINFORCE algorithm (Williams, 1992):

$$\nabla_\theta J(\theta \mid s) = \mathbb{E}_{\pi \sim p_\theta(.|s)} \left[ \big( L(\pi \mid s) - b(s) \big) \nabla_\theta \log p_\theta(\pi \mid s) \right] , \tag{4}$$

where $b(s)$ denotes a baseline function that does not depend on $\pi$ and estimates the expected tour length to reduce the variance of the gradients.

By drawing $B$ *i.i.d.* sample graphs $s_1, s_2, \dots, s_B \sim \mathcal{S}$ and sampling a single tour per graph, *i.e.* $\pi_i \sim p_\theta(. \mid s_i)$, the gradient in (4) is approximated with Monte Carlo sampling as follows:

$$\nabla_\theta J(\theta) \approx \frac{1}{B} \sum_{i=1}^{B} \Big( L(\pi_i|s_i) - b(s_i) \Big) \nabla_\theta \log p_\theta(\pi_i \mid s_i) . \tag{5}$$

A simple and popular choice of the baseline $b(s)$ is an exponential moving average of the rewards obtained by the network over time to account for the fact that the policy improves with training. While this choice of baseline proved sufficient to improve upon the Christofides algorithm, it suffers from not being able to differentiate between different input graphs. In particular, the optimal tour $\pi^*$ for a difficult graph $s$ may be still discouraged if $L(\pi^*|s) > b$ because $b$ is shared across all instances in the batch.

Using a parametric baseline to estimate the expected tour length $\mathbb{E}_{\pi \sim p_\theta(.|s)} L(\pi \mid s)$ typically improves learning. Therefore, we introduce an auxiliary network, called a *critic* and parameterized

---

**Algorithm 2** Active Search

1: **procedure** ACTIVESEARCH(input s, $\theta$, number of candidates K, B, $\alpha$)
2: $\pi \leftarrow$ RANDOMSOLUTION()
3: $L_\pi \leftarrow L(\pi \mid s)$
4: $n \leftarrow \lceil \frac{K}{B} \rceil$
5: **for** $t = 1 \ldots n$ **do**
6: $\pi_i \sim$ SAMPLESOLUTION$(p_\theta(. \mid s))$ for $i \in \{1, \ldots, B\}$
7: $j \leftarrow$ ARGMIN$(L(\pi_1 \mid s) \ldots L(\pi_B \mid s))$
8: $L_j \leftarrow L(\pi_j \mid s)$
9: **if** $L_j < L_\pi$ **then**
10: $\pi \leftarrow \pi_j$
11: $L_\pi \leftarrow L_j$
12: **end if**
13: $g_\theta \leftarrow \frac{1}{B} \sum_{i=1}^{B} (L(\pi_i \mid s) - b) \nabla_\theta \log p_\theta(\pi_i \mid s)$
14: $\theta \leftarrow$ ADAM$(\theta, g_\theta)$
15: $b \leftarrow \alpha \times b + (1 - \alpha) \times (\frac{1}{B} \sum_{i=1}^{B} b_i)$
16: **end for**
17: **return** $\pi$
18: **end procedure**

---

by $\theta_v$, to learn the expected tour length found by our current policy $p_\theta$ given an input sequence $s$. The critic is trained with stochastic gradient descent on a mean squared error objective between its predictions $b_{\theta_v}(s)$ and the actual tour lengths sampled by the most recent policy. The additional objective is formulated as

$$\mathcal{L}(\theta_v) = \frac{1}{B} \sum_{i=1}^{B} \left\| b_{\theta_v}(s_i) - L(\pi_i \mid s_i) \right\|_2^2 . \tag{6}$$

**Critic's architecture for TSP.** We now explain how our critic maps an input sequence $s$ into a baseline prediction $b_{\theta_v}(s)$. Our critic comprises three neural network modules: 1) an LSTM encoder, 2) an LSTM process block and 3) a 2-layer ReLU neural network decoder. Its encoder has the same architecture as that of our pointer network's encoder and encodes an input sequence $s$ into a sequence of latent memory states and a hidden state $h$. The process block, similarly to (Vinyals et al., 2015a), then performs P steps of computation over the hidden state $h$. Each processing step updates this hidden state by glimpsing at the memory states as described in Appendix A.1 and feeds the output of the glimpse function as input to the next processing step. At the end of the process block, the obtained hidden state is then decoded into a baseline prediction (i.e a single scalar) by two fully connected layers with respectively d and 1 unit(s).

Our training algorithm, described in Algorithm 1, is closely related to the asynchronous advantage actor-critic (A3C) proposed in (Mnih et al., 2016), as the difference between the sampled tour lengths and the critic's predictions is an unbiased estimate of the advantage function. We perform our updates asynchronously across multiple workers, but each worker also handles a mini-batch of graphs for better gradient estimates.

### 4.1 SEARCH STRATEGIES

As evaluating a tour length is inexpensive, our TSP agent can easily simulate a search procedure at inference time by considering multiple candidate solutions per graph and selecting the best. This inference process resembles how solvers search over a large set of feasible solutions. In this paper, we consider two search strategies detailed below, which we refer to as *sampling* and *active search*.

**Sampling.** Our first approach is simply to sample multiple candidate tours from our stochastic policy $p_\theta(.|s)$ and select the shortest one. In contrast to heuristic solvers, we do not enforce our model to sample different tours during the process. However, we can control the diversity of the sampled tours with a temperature hyperparameter when sampling from our non-parametric softmax (see Appendix A.2). This sampling process yields significant improvements over greedy decoding, which always selects the index with the largest probability. We also considered perturbing the pointing

Table 1: Different learning configurations.

| Configuration | Learn on training data | Sampling on test set | Refining on test set |
|---|---|---|---|
| RL pretraining-Greedy | Yes | No | No |
| Active Search (AS) | No | Yes | Yes |
| RL pretraining-Sampling | Yes | Yes | No |
| RL pretraining-Active Search | Yes | Yes | Yes |

mechanism with random noise and greedily decoding from the obtained modified policy, similarly to (Cho, 2016), but this proves less effective than sampling in our experiments.

**Active Search.**  Rather than sampling with a fixed model and ignoring the reward information obtained from the sampled solutions, one can refine the parameters of the stochastic policy $p_\theta$ during inference to minimize $\mathbb{E}_{\pi \sim p_\theta(.|s)} L(\pi \mid s)$ on a *single test input s*. This approach proves especially competitive when starting from a trained model. Remarkably, it also produces satisfying solutions when starting from an untrained model. We refer to these two approaches as *RL pretraining-Active Search* and *Active Search* because the model actively updates its parameters while searching for candidate solutions on a single test instance.

Active Search applies policy gradients similarly to Algorithm 1 but draws Monte Carlo samples over candidate solutions $\pi_1 \ldots \pi_B \sim p_\theta(\cdot|s)$ for a single test input. It resorts to an exponential moving average baseline, rather than a critic, as there is no need to differentiate between inputs. Our Active Search training algorithm is presented in Algorithm 2. We note that while RL training does not require supervision, it still requires training data and hence generalization depends on the training data distribution. In contrast, Active Search is distribution independent. Finally, since we encode a set of cities as a sequence, we randomly shuffle the input sequence before feeding it to our pointer network. This increases the stochasticity of the sampling procedure and leads to large improvements in Active Search.

## 5    EXPERIMENTS

We conduct experiments to investigate the behavior of the proposed Neural Combinatorial Optimization methods. We consider three benchmark tasks, Euclidean TSP20, 50 and 100, for which we generate a test set of $1,000$ graphs. Points are drawn uniformly at random in the unit square $[0, 1]^2$.

### 5.1    EXPERIMENTAL DETAILS

Across all experiments, we use mini-batches of 128 sequences, LSTM cells with 128 hidden units, and embed the two coordinates of each point in a 128-dimensional space. We train our models with the Adam optimizer (Kingma & Ba, 2014) and use an initial learning rate of $10^{-3}$ for TSP20 and TSP50 and $10^{-4}$ for TSP100 that we decay every 5000 steps by a factor of 0.96. We initialize our parameters uniformly at random within $[-0.08, 0.08]$ and clip the $L2$ norm of our gradients to 1.0. We use up to one attention glimpse. When searching, the mini-batches either consist of replications of the test sequence or its permutations. The baseline decay is set to $\alpha = 0.99$ in Active Search. Our model and training code in Tensorflow (Abadi et al., 2016) will be made availabe soon. Table 1 summarizes the configurations and different search strategies used in the experiments. The variations of our method, experimental procedure and results are as follows.

**Supervised Learning.**  In addition to the described baselines, we implement and train a pointer network with supervised learning, similarly to (Vinyals et al., 2015b). While our supervised data consists of one million optimal tours, we find that our supervised learning results are not as good as those reported in by (Vinyals et al., 2015b). We suspect that learning from optimal tours is harder for supervised pointer networks due to subtle features that the model cannot figure out only by looking at given supervised targets. We thus refer to the results in (Vinyals et al., 2015b) for TSP20 and TSP50 and report our results on TSP100, all of which are suboptimal compared to other approaches.

Table 2: Average tour lengths (lower is better). Results marked [†] are from (Vinyals et al., 2015b).

| Task | Supervised Learning | RL pretraining | | | | AS | Christo-fides | OR Tools' local search | Optimal |
|---|---|---|---|---|---|---|---|---|---|
| | | greedy | greedy@16 | sampling | AS | | | | |
| TSP20 | 3.88[†] | 3.89 | – | 3.82 | 3.82 | 3.96 | 4.30 | 3.85 | 3.82 |
| TSP50 | 6.09[†] | 5.95 | 5.80 | 5.70 | 5.70 | 5.87 | 6.62 | 5.80 | 5.68 |
| TSP100 | 10.81 | 8.30 | 7.97 | 7.88 | 7.83 | 8.19 | 9.18 | 7.99 | 7.77 |

**RL pretraining.**    For the RL experiments, we generate training mini-batches of inputs on the fly and update the model parameters with the Actor Critic Algorithm 1. We use a validation set of $10,000$ randomly generated instances for hyper-parameters tuning. Our critic consists of an encoder network which has the same architecture as that of the policy network, but followed by 3 processing steps and 2 fully connected layers. We find that clipping the logits to $[-10, 10]$ with a $\tanh(\cdot)$ activation function, as described in Appendix A.2, helps with exploration and yields marginal performance gains. The simplest search strategy using an RL pretrained model is greedy decoding, *i.e.* selecting the city with the largest probability at each decoding step. We also experiment with decoding greedily from a set of 16 pretrained models at inference time. For each graph, the tour found by each individual model is collected and the shortest tour is chosen. We refer to those approaches as *RL pretraining-greedy* and *RL pretraining-greedy@16*.

**RL pretraining-Sampling.**    For each test instance, we sample $1,280,000$ candidate solutions from a pretrained model and keep track of the shortest tour. A grid search over the temperature hyperparameter found respective temperatures of 2.0, 2.2 and 1.5 to yield the best results for TSP20, TSP50 and TSP100. We refer to the tuned temperature hyperparameter as $T^*$. Since sampling does not require parameter udpates and is entirely parallelizable, we use a larger batch size for speed purposes.

**RL pretraining-Active Search.**    For each test instance, we initialize the model parameters from a pretrained RL model and run Active Search for up to $10,000$ training steps with a batch size of 128, sampling a total of $1,280,000$ candidate solutions. We set the learning rate to a hundredth of the initial learning rate the TSP agent was trained on (i.e. $10^{-5}$ for TSP20/TSP50 and $10^{-6}$ for TSP100).

**Active Search.**    We allow the model to train much longer to account for the fact that it starts from scratch. For each test graph, we run Active Search for $100,000$ training steps on TSP20/TSP50 and $200,000$ training steps on TSP100.

## 5.2    RESULTS AND ANALYSES

We compare our methods against 3 different baselines of increasing performance and complexity: 1) Christofides, 2) the vehicle routing solver from OR-Tools (Google, 2016) and 3) optimality. Christofides solutions are obtained in polynomial time and guaranteed to be within a 1.5 ratio of optimality. OR-Tools improves over Christofides' solutions with simple local search operators, including 2-opt (Johnson, 1990) and a version of the Lin-Kernighan heuristic (Lin & Kernighan, 1973), stopping when it reaches a local minimum. In order to escape poor local optima, OR-Tools' local search can also be run in conjunction with different metaheuristics, such as simulated annealing (Kirkpatrick et al., 1983), tabu search (Glover & Laguna, 2013) or guided local search (Voudouris & Tsang, 1999). OR-Tools' vehicle routing solver can tackle a superset of the TSP and operates at a higher level of generality than solvers that are highly specific to the TSP. While not state-of-the art for the TSP, it is a common choice for general routing problems and provides a reasonable baseline between the simplicity of the most basic local search operators and the sophistication of the strongest solvers. Optimal solutions are obtained via Concorde (Applegate et al., 2006) and LK-H's local search  (Helsgaun, 2012; 2000). While only Concorde provably solves instances to optimality, we empirically find that LK-H also achieves optimal solutions on all of our test sets after 50 trials per graph (which is the default parameter setting).

We report the average tour lengths of our approaches on TSP20, TSP50, and TSP100 in Table 2. Notably, results demonstrate that training with RL significantly improves over supervised learning

Table 3: Running times in seconds (s) of greedy methods compared to OR Tool's local search and solvers that find the optimal solutions. Time is measured over the entire test set and averaged. LK-H was run for 50 trials per graph (the default parameter setting). It is likely that optimal solutions were found in fewer trials, resulting in shorter running times.

| Task | RL pretraining | | OR-Tools' | Optimal | |
|---|---|---|---|---|---|
| | greedy | greedy@16 | local search | Concorde | LK-H |
| TSP50 | 0.003s | 0.04s | 0.02s | 0.05s | 0.14s |
| TSP100 | 0.01s | 0.15s | 0.10s | 0.22s | 0.88s |

Table 4: Average tour lengths of RL pretraining-Sampling and RL pretraining-Active Search as they sample more solutions. Corresponding running times on a single Tesla K80 GPU are in parantheses.

| Task | # Solutions | RL pretraining | | |
|---|---|---|---|---|
| | | Sampling $T = 1$ | Sampling $T = T^*$ | Active Search |
| TSP50 | 128 | 5.80 (3.4s) | 5.80 (3.4s) | 5.80 (0.5s) |
| | 1,280 | 5.77 (3.4s) | 5.75 (3.4s) | 5.76 (5s) |
| | 12,800 | 5.75 (13.8s) | 5.73 (13.8s) | 5.74 (50s) |
| | 128,000 | 5.73 (110s) | 5.71 (110s) | 5.72 (500s) |
| | 1,280,000 | 5.72 (1080s) | 5.70 (1080s) | 5.70 (5000s) |
| TSP100 | 128 | 8.05 (10.3s) | 8.09 (10.3s) | 8.04 (1.2s) |
| | 1,280 | 8.00 (10.3s) | 8.00 (10.3s) | 7.98 (12s) |
| | 12,800 | 7.95 (31s) | 7.95 (31s) | 7.92 (120s) |
| | 128,000 | 7.92 (265s) | 7.91 (265s) | 7.87 (1200s) |
| | 1,280,000 | 7.89 (2640s) | 7.88 (2640s) | 7.83 (12000s) |

(Vinyals et al., 2015b). All our methods comfortably surpass Christofides' heuristic, including RL pretraining-Greedy which also does not rely on search. Table 3 compares the running times of our greedy methods to the aforementioned baselines, with our methods running on a single Nvidia Tesla K80 GPU, Concorde and LK-H running on an Intel Xeon CPU E5-1650 v3 3.50GHz CPU and OR-Tool on an Intel Haswell CPU. We find that both greedy approaches are time-efficient but still quite far from optimality.

Searching at inference time proves crucial to get closer to optimality but comes at the expense of longer running times. Fortunately, the search from RL pretraining-Sampling and RL pretraining-Active Search can be stopped early with a small performance tradeoff in terms of the final objective. This can be seen in Table 4, where we show their performances and corresponding running times as a function of how many solutions they consider.

We also find that many of our RL pretraining methods outperform OR-Tools' local search, including RL pretraining-Greedy@16 which runs similarly fast. Table 6 in Appendix A.3 presents the performance of the metaheuristics as they consider more solutions and the corresponding running times. In our experiments, Neural Combinatorial proves superior than Simulated Annealing but is slightly less competitive that Tabu Search and much less so than Guided Local Search.

We present a more detailed comparison of our methods in Figure 3, where we sort the ratios to optimality of our different learning configurations. RL pretraining-Sampling and RL pretraining-Active Search are the most competitive Neural Combinatorial Optimization methods and recover the optimal solution in a significant number of our test cases. We find that for small solution spaces, RL pretraining-Sampling, with a finetuned softmax temperature, outperforms RL pretraining-Active Search with the latter sometimes orienting the search towards suboptimal regions of the solution space (see TSP50 results in Table 4 and Figure 3). Furthermore, RL pretraining-Sampling benefits from being fully parallelizable and runs faster than RL pretraining-Active Search. However, for larger solution spaces, RL-pretraining Active Search proves superior both when controlling for the number of sampled solutions or the running time. Interestingly, Active Search - which starts from an untrained model - also produces competitive tours but requires a considerable amount of time (respectively 7 and 25 hours per instance of TSP50/TSP100). Finally, we show randomly picked example tours found by our methods in Figure 4 in Appendix A.4.

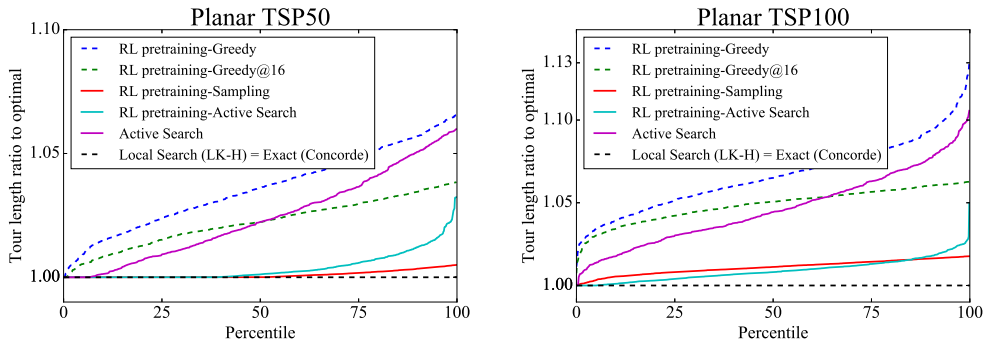

Figure 3: Sorted tour length ratios to optimality

# 6 GENERALIZATION TO OTHER PROBLEMS

In this section, we discuss how to apply Neural Combinatorial Optimization to other problems than the TSP. In Neural Combinatorial Optimization, the model architecture is tied to the given combinatorial optimization problem. Examples of useful networks include the pointer network, when the output is a permutation or a truncated permutation or a subset of the input, and the classical seq2seq model for other kinds of structured outputs. For combinatorial problems that require to assign labels to elements of the input, such as graph coloring, it is also possible to combine a pointer module and a softmax module to simultaneously point and assign at decoding time. Given a model that encodes an instance of a given combinatorial optimization task and repeatedly branches into subtrees to construct a solution, the training procedures described in Section 4 can then be applied by adapting the reward function depending on the optimization problem being considered.

Additionally, one also needs to ensure the feasibility of the obtained solutions. For certain combinatorial problems, it is straightforward to know exactly which branches do not lead to any feasible solutions at decoding time. We can then simply manually assign them a zero probability when decoding, similarly to how we enforce our model to not point at the same city twice in our pointing mechanism (see Appendix A.1). However, for many combinatorial problems, coming up with a feasible solution can be a challenge in itself. Consider, for example, the Travelling Salesman Problem with Time Windows, where the travelling salesman has the additional constraint of visiting each city during a specific time window. It might be that most branches being considered early in the tour do not lead to any solution that respects all time windows. In such cases, knowing exactly which branches are feasible requires searching their subtrees, a time-consuming process that is not much easier than directly searching for the optimal solution unless using problem-specific heuristics.

Rather than explicitly constraining the model to only sample feasible solutions, one can also let the model learn to respect the problem's constraints. A simple approach, to be verified experimentally in future work, consists in augmenting the objective function with a term that penalizes solutions for violating the problem's constraints, similarly to penalty methods in constrained optimization. While this does not guarantee that the model consistently samples feasible solutions at inference time, this is not necessarily problematic as we can simply ignore infeasible solutions and resample from the model (for RL pretraining-Sampling and RL-pretraining Active Search). It is also conceivable to combine both approaches by assigning zero probabilities to branches that are easily identifiable as infeasible while still penalizing infeasible solutions once they are entirely constructed.

## 6.1 KNAPSACK EXAMPLE

As an example of the flexibility of Neural Combinatorial Optimization, we consider the KnapSack problem, another intensively studied problem in computer science. Given a set of $n$ items $i = 1...n$, each with weight $w_i$ and value $v_i$ and a maximum weight capacity of $W$, the 0-1 KnapSack problem consists in maximizing the sum of the values of items present in the knapsack so that the sum of the

weights is less than or equal to the knapsack capacity:

$$\max_{S \subseteq \{1,2,\ldots,n\}} \quad \sum_{i \in S} v_i$$
$$\text{subject to} \quad \sum_{i \in S} w_i \leq W \tag{7}$$

With $w_i$, $v_i$ and $W$ taking real values, the problem is NP-hard (Kellerer et al., 2004). A naive heuristic is to take the items ordered by their weight-to-value ratios until they fill up the weight capacity. Two simple heuristics are ExpKnap, which employs branch-and-bound with Linear Programming bounds (Pisinger, 1995), and MinKnap, which uses dynamic programming with enumerative bounds (Pisinger, 1997). Exact solutions can also be obtained by quantizing the weights to high precisions and then performing dynamic programming with pseudo-polynomial complexity (Bertsimas & Demir, 2002).

We apply the pointer network and encode each KnapSack instance as a sequence of 2D vectors $(w_i, v_i)$. At decoding time, the pointer network points to items to include in the knapsack and stops when the total weight of the items collected so far exceeds the weight capacity. We generate three datasets, KNAP50, KNAP100 and KNAP200, of a thousand instances with items' weights and values drawn uniformly at random in $[0, 1]$. Without loss of generality (since we can scale the items' weights), we set the capacities to 12.5 for KNAP50 and 25 for KNAP100 and KNAP200. We present the performances of RL pretraining-Greedy and Active Search (which we run for $5,000$ training steps) in Table 5 and compare them to the following baselines: 1) random search (which we let sample as many feasible solutions seen by Active Search), 2) the greedy value-to-weight ratio heuristic, 3) MinKnap, 4) ExpKnap, 5) OR-Tools' KnapSack solver (Google, 2016) and 6) optimality (which we obtained by quantizing the weights to high precisions and using dynamic programming).

Table 5: Results of RL pretraining-Greedy and Active Search on KnapSack (higher is better).

| Task | RL pretraining greedy | Active Search | Random Search | Greedy | MinKnap / ExpKnap / OR-Tools | Optimal |
|---|---|---|---|---|---|---|
| KNAP50 | 19.86 | **20.07** | 17.91 | 19.24 | **20.07** | **20.07** |
| KNAP100 | 40.27 | **40.50** | 33.23 | 38.53 | **40.50** | **40.50** |
| KNAP200 | 57.10 | **57.45** | 35.95 | 55.42 | **57.45** | **57.45** |

## 7 CONCLUSION

This paper presents Neural Combinatorial Optimization, a framework to tackle combinatorial optimization with reinforcement learning and neural networks. We focus on the traveling salesman problem (TSP) and present a set of results for each variation of the framework. Experiments demonstrate that Neural Combinatorial Optimization achieves close to optimal results on 2D Euclidean graphs with up to 100 nodes. Our results, while still far from the strongest solvers (especially those which are optimized for one problem), provide an interesting research avenue for using neural networks as a general tool for tackling combinatorial optimization problems.

## ACKNOWLEDGMENTS

The authors would like to thank Vincent Furnon, Mustafa Ispir, Lukasz Kaiser, Oriol Vinyals, Barret Zoph, the Google Brain team and the anonymous ICLR reviewers for insightful comments and discussion.

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

# A  APPENDIX

## A.1  POINTING AND ATTENDING

**Pointing mechanism:**  Its computations are parameterized by two attention matrices $W_{ref}, W_q \in \mathbb{R}^{d \times d}$ and an attention vector $v \in \mathbb{R}^d$ as follows:

$$u_i = \begin{cases} v^\top \cdot \tanh\left(W_{ref} \cdot r_i + W_q \cdot q\right) & \text{if } i \neq \pi(j) \text{ for all } j < i \\ -\infty & \text{otherwise} \end{cases} \text{ for } i = 1, 2, ..., k \qquad (8)$$

$$A(ref, q; W_{ref}, W_q, v) \stackrel{\text{def}}{=} softmax(u). \qquad (9)$$

Our pointer network, at decoder step $j$, then assigns the probability of visiting the next point $\pi(j)$ of the tour as follows:

$$p(\pi(j)|\pi(<j), s) \stackrel{\text{def}}{=} A(enc_{1:n}, dec_j). \qquad (10)$$

Setting the logits of cities that already appeared in the tour to $-\infty$, as shown in Equation 8, ensures that our model only points at cities that have yet to be visited and hence outputs valid TSP tours.

**Attending mechanism:**  Specifically, our glimpse function $G(ref, q)$ takes the same inputs as the attention function $A$ and is parameterized by $W_{ref}^g, W_q^g \in \mathbb{R}^{d \times d}$ and $v^g \in \mathbb{R}^d$. It performs the following computations:

$$p = A(ref, q; W_{ref}^g, W_q^g, v^g) \qquad (11)$$

$$G(ref, q; W_{ref}^g, W_q^g, v^g) \stackrel{\text{def}}{=} \sum_{i=1}^{k} r_i p_i. \qquad (12)$$

The glimpse function $G$ essentially computes a linear combination of the reference vectors weighted by the attention probabilities. It can also be applied multiple times on the same reference set $ref$:

$$g_0 \stackrel{\text{def}}{=} q \qquad (13)$$

$$g_l \stackrel{\text{def}}{=} G(ref, g_{l-1}; W_{ref}^g, W_q^g, v^g) \qquad (14)$$

Finally, the ultimate $g_l$ vector is passed to the attention function $A(ref, g_l; W_{ref}, W_q, v)$ to produce the probabilities of the pointing mechanism. We observed empirically that glimpsing more than once with the same parameters made the model less likely to learn and barely improved the results.

## A.2  IMPROVING EXPLORATION

**Softmax temperature:**  We modify Equation 9 as follows:

$$A(ref, q, T; W_{ref}, W_q, v) \stackrel{\text{def}}{=} softmax(u/T), \qquad (15)$$

where $T$ is a *temperature* hyperparameter set to $T = 1$ during training. When $T > 1$, the distribution represented by $A(ref, q)$ becomes less steep, hence preventing the model from being overconfident.

**Logit clipping:**  We modify Equation 9 as follows:

$$A(ref, q; W_{ref}, W_q, v) \stackrel{\text{def}}{=} softmax(C \tanh(u)), \qquad (16)$$

where $C$ is a hyperparameter that controls the range of the logits and hence the entropy of $A(ref, q)$.

## A.3   OR TOOL'S METAHEURISTICS BASELINES FOR TSP

Table 6: Performance of OR-Tools' metaheuristics as they consider more solutions. Corresponding running times in seconds (s) on a single Intel Haswell CPU are in parantheses.

| Task | #Solutions | Simulated Annealing | Tabu Search | Guided Local Search |
|---|---|---|---|---|
| TSP50 | 1 | 6.62 (0.03s) | 6.62 (0.03s) | 6.62 (0.03s) |
| | 128 | 5.81 (0.24s) | 5.79 (3.4s) | 5.76 (0.5s) |
| | 1,280 | 5.81 (4.2s) | 5.73 (36s) | 5.69 (5s) |
| | 12,800 | 5.81 (44s) | 5.69 (330s) | 5.68 (48s) |
| | 128,000 | 5.81 (460s) | 5.68 (3200s) | 5.68 (450s) |
| | 1,280,000 | 5.81 (3960s) | 5.68 (29650s) | 5.68 (4530s) |
| TSP100 | 1 | 9.18 (0.07s) | 9.18 (0.07s) | 9.18 (0.07s) |
| | 128 | 8.00 (0.67s) | 7.99 (15.3s) | 7.94 (1.44s) |
| | 1,280 | 7.99 (15.7s) | 7.93 (255s) | 7.84 (18.4s) |
| | 12,800 | 7.99 (166s) | 7.84 (2460s) | 7.77 (182s) |
| | 128,000 | 7.99 (1650s) | 7.79 (22740s) | 7.77 (1740s) |
| | 1,280,000 | 7.99 (15810s) | 7.78 (208230s) | 7.77 (16150s) |

## A.4   SAMPLE TOURS

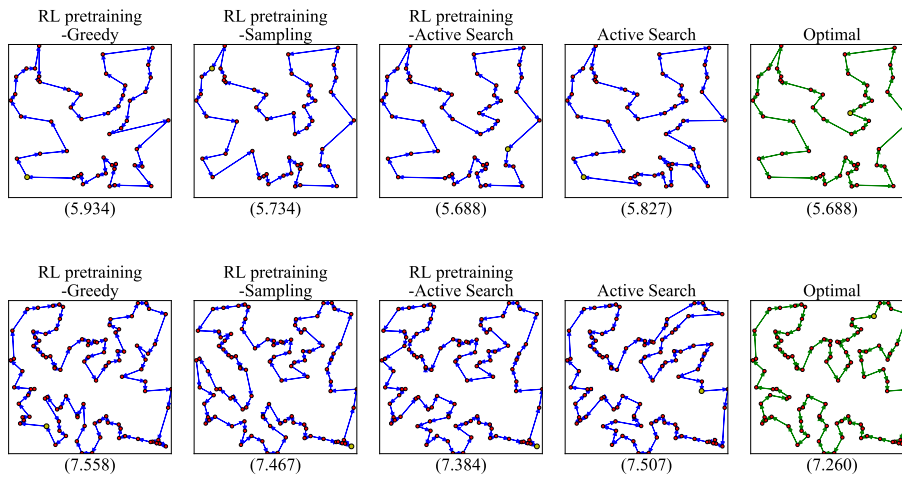

Figure 4: Sample tours. Top: TSP50; Bottom: TSP100.

