# Peer review of "Neural Combinatorial Optimization with Reinforcement Learning"

_ICLR 2017 — rejected_

[Public Comment · (anonymous) · 07 Nov 2016]
**Related work incomplete**

There is a large body of work on solving TSP instances that this paper ignores. In particular, the concorde algorithm has produced provably optimal solutions to problems as large as 85,900 cities, and can solve 100+ city problems in a few seconds on a single 500MHz core. Thus, the claims made that this is even close to being a useful tool for solving TSP problems are demonstrably untrue.

[Reviewer Comment · AnonReviewer3 · 03 Dec 2016]
**Code availability**

I am very glad to read "Our model and training code will be made available soon." Thanks for that! My question is: how soon is soon? During the review period? In time for the conference?

[Reviewer Comment · AnonReviewer3 · 03 Dec 2016]
**Is RL pretraining Sampling T=T* actually better than RL pretraining AS when compared with the same number of 10.000 batches?**

In Table 3, what is the performance for the missing values of RL pretraining with 10.000 batches for Sampling T=1 and T=T*? 

Since performance improved much more from 100 to 1.000 batches for RL pretraining Sampling T=T* than it did for RL pretraining AS (e.g., 5.79->5.71 vs 5.74->5.71 for TSP50), I would expect RL pretraining Sampling T=T* to do better than RL pretraining AS when you use 10.000 samples. This would also change your qualitative conclusion in Table 2 and the overall result of the paper. You seem to glance over this in the text by saying "we sample 1000 batches from a pretrained model, afer which we do not see significant improvement", but seeing the much larger "gradient" from 50, 100, and 1000 batches than for RL pretraining AS, and seeing how key the result is to the final take-away from the paper, I would be far more convinced by just seeing the numbers for 10.000 batches.

Also, what is actually the difference between RL pretraining Sampling T=1 and T=T*? (Maybe I just missed this in the text.)

[Official Review · AnonReviewer2 · rating 6 · confidence 4 · 15 Dec 2016]
**No Title**

This paper proposes to use RNN and reinforcement learning for solving combinatorial optimization problems. The use of pointer network is interesting as it enables generalization to arbitrary input size. The proposed method also "fintunes" on test examples with active search to achieve better performance.

The proposed method is theoretically interesting as it shows that RNN and RL can be combined to solve combinatorial optimization problems and achieve comparable performance to traditional heuristic based algorithms.

However, the lack of complexity comparison against baselines make it impossible to tell whether the proposed method has any practical value. The matter is further complicated by the fact that the proposed method runs on GPU while baselines run on CPU: it is hard to even come up with a meaningful unit of complexity. Money spent on hardware and electricity per instance may be a viable option.

Further more, the performance comparisons should be taken with a grain of salt as traditional heuristic based algorithms can often give better performance if allowed more computation, which is not controlled across algorithms.

[Public Comment · (anonymous) · 15 Dec 2016]
**About Concorde**

This is very interesting to me! Thank you for this.

After reading this paper, I tested the Concorde. I think the Concorde allows only integer distances(if use Euclidean distance, they round off), so cannot provide optimal solution of Euclidean TSP.
But error can be small if multiply the distance by a large constant.

I want to know that, if I correct, does 'optimal' means a solution which is very closed to optimal?

[Official Review · AnonReviewer3 · rating 6 · confidence 4 · 16 Dec 2016 (modified: 19 Jan 2017)]
**Promising method, but biased presentation**

This paper is methodologically very interesting, and just based on the methodological contribution I would vote for acceptance. However, the paper's sweeping claims of clearly beating existing baselines for TSP have been shown to not hold, with the local search method LK-H solving all the authors' instances to optimality -- in seconds on a CPU, compared to clearly suboptimal results by the authors' method in 25h on a GPU. 

Seeing this clear dominance of the local search method LK-H, I find it irresponsible by the authors that they left Figure 1 as it is -- with the line for "local search" referring to an obviously poor implementation by Google rather than the LK-H local search method that everyone uses. For example, at NIPS, I saw this Figure 1 being used in a talk (I am not sure anymore by whom, but I don't think it was by the authors), the narrative being "RNNs now also clearly perform better than local search". Of course, people would use a figure like that for that purpose, and it is clearly up to the authors to avoid such misconceptions. 

The right course of action upon realizing the real strength of local search with LK-H would've been to make "local search" the same line as "Optimal", showing that the authors' method is still far worse than proper local search. But the authors chose to leave the figure as it was, still suggesting that their method is far better than local search. Probably the authors didn't even think about this, but this of course will mislead the many superficial readers. To people outside of deep learning, this must look like a sensational yet obviously wrong claim. I thus vote for rejection despite the interesting method. 

------------------------

Update after rebuttal and changes:

I'm torn about this paper. 

On the one hand, the paper is very well written and I do think the method is very interesting and promising. I'd even like to try it and improve it in the future. So, from that point of view a clear accept.

On the other hand, the paper was using extremely poor baselines, making the authors' method appear sensationally strong in comparison, and over the course of many iterations of reviewer questions and anonymous feedback, this has come down to the authors' methods being far inferior to the state of the art. That's fine (I expected that all along), but the problem is that the authors don't seem to want this to be true... E.g., they make statements, such as "We find that both greedy approaches are time-efficient and just a few percents worse than optimality."
That statement may be true, but it is very well known in the TSP community that it is typically quite trivial to get to a few percent worse than optimality. What's hard and interesting is to push those last few percent. 
(As a side note: the authors probably don't stop LK-H once it has found the optimal solution, like they do with their own method after finding a local optimum. LK-H is an anytime algorithm, so even if it ran for a day that doesn't mean that it didn't find the optimal solution after milliseconds -- and a solution a few percent suboptimal even faster).

Nevertheless, since the claims have been toned down over the course of the many iterations, I was starting to feel more positive about this paper when just re-reading it. That is, until I got to the section on Knapsack solving. The version of the paper I reviewed was not bad here, as it at least stated two simple heuristics that yield optimal solutions:

"Two simple heuristics are ExpKnap, which employs brand-and-bound with Linear Programming bounds (Pisinger, 1995), and MinKnap, which employs dynamic programming with enumerative bounds (Pisinger, 1997). Exact solutions can also be optained by quantizing the weights to high precisions and then performing dynamic programming with a pseudo-polynomial complexity (Bertsimas & Demir, 2002)." That version then went on to show that these simple heuristics were already optimal, just like their own method.

In a revision between December 11 and 14, however, that paragraph, along with the optimal results of ExpKnap and MinKnap seems to have been dropped, and the authors instead introduced two new poor baseline methods (random search and greedy). This was likely in an effort to find some methods that are not optimal on these very easy instances. I personally find it pointless to present results for random search here, as nobody would use that for TSP. It's like comparing results on MNIST against a decision stump (yes, you'll do better than that, but that is not surprising). The results for greedy are interesting to see. However, dropping the strong results of the simple heuristics ExpKnap and MinKnap (and their entire discussion) appears unresponsible, since the resulting table in the new version of the paper now suggests that the authors' method is better than all baselines. Of course, if all that one is after is a column of bold numbers for ones own approach that's what one can do, but I don't find it responsible to hide the better baselines. Also, why don't the authors try at least the same OR-tools solver from Google that they tried for TSP? It seems to support Knapsack directly:

[Official Review · AnonReviewer4 · rating 6 · confidence 4 · 27 Dec 2016]
**Promising approach to combinatorial optimization**

This paper applies the pointer network architecture—wherein an attention mechanism is fashioned to point to elements of an input sequence, allowing a decoder to output said elements—in order to solve simple combinatorial optimization problems such as the well-known travelling salesman problem. The network is trained by reinforcement learning using an actor-critic method, with the actor trained using the REINFORCE method, and the critic used to estimate the reward baseline within the REINFORCE objective.

The paper is well written and easy to understand. Its use of a reinforcement learning and attention model framework to learn the structure of the space in which combinatorial problems of variable size can be tackled appears novel. Importantly, it provides an interesting research avenue for revisiting classical neural-based solutions to some combinatorial optimization problems, using recently-developed sequence-to-sequence approaches. As such, I think it merits consideration for the conference.

I have a few comments and some important reservations with the paper:

1) I take exception to the conclusion that the pointer network approach can handle general types of combinatorial optimization problems. The crux of combinatorial problems — for practical applications — lies in the complex constraints that define feasible solutions (e.g. simple generalizations of the TSP that involve time windows, or multiple salesmen). For these problems, it is no longer so simple to exclude possible solutions from the enumeration of the solution by just « striking off » previously-visited instances; in fact, for many of these problems, finding a single feasible solution might in general be a challenge. It would be relevant to include a discussion of whether the Neural Combinatorial Optimization approach could scale to these important classes of problems, and if so, how. My understanding is that this approach, as presented, would be mostly suitable for assignment problems with a very simple constraint structure.

2) The operations research literature is replete with a large number of benchmark problems that have become standard to compare solver quality. For instance, TSPLIB contains a large number of TSP instances (

[Author Response · Irwan Bello · 05 Jan 2017]
**Summary of paper's revision**

We thank reviewers for their valuable feedback that helped us improve the paper. We appreciate their interest in the method and its novelty. We have made several changes to the paper which are summarized below. We ask reviewers to evaluate the new version of the paper and adjust their reviews if necessary.

1) Previous Figure 1, which was problematic due to different possible interpretations of “local search” was removed.

2) We added precise running time evaluations for all of the methods in the paper. Table 3 presents running time of the RL pretraining-greedy method and the solvers we compare against. Table 4 presents the performance and corresponding running time of RL pretraining-Sampling and RL pretraining-Active Search as a function of the number solutions considered. It shows how they can be stopped early at the cost of a small performance degradation. Table 6 contains the same information for the metaheuristics from OR-Tools vehicle routing library solver. We controlled the complexity of these approaches by letting all of them evaluate 1,280,000 solutions. Section 5.2 was rewritten in light of the new results.

3) We experimented with a new approach, called RL pretraining-Greedy@16, that decodes greedily from 16 different pretrained models at inference time and selects the shortest tour. It runs as fast as the solvers while only suffering from a small performance cost.

4) We added a discussion in Section 6 (Generalization to other problems) explaining how one may apply Neural Combinatorial Optimization to problems for which coming up with a feasible solution is challenging by itself.

5) We added a more detailed description of the critic network (see Section 4 - Critic’s architecture for TSP).

Please take a look and let us know your thoughts.

[Public Comment · (anonymous) · 06 Jan 2017]
**Question**

I posted this question in a response below, but it seems to be getting ignored so I thought I'd bring it to the top, with some additional points.

Thanks for the update. The natural question to ask, then is - do there exist many (or any) problems that are both interesting and have not been, and cannot be, addressed by the existing combinatorial optimization community? You knock existing algorithms for being "highly optimized" to particular problems, but if every worthwhile problem has "highly optimized" solutions, what good is your work? 

Also, please stop calling existing TSP solvers such as concorde a heuristic. Concorde produces solutions which are provably correct. Your approach does not, nor is it remotely close. From a practical perspective, this is an important distinction; I don't see why anyone would choose the latter when given the choice. The second paragraphs of the related work and introduction are guilty of this. Also in the related work - you say it solves cities with "thousands of cities" when it has solved a 85k problem. 

I'd also echo concerns about the toy-ness of the evaluation metrics here - 100 cities is 800x smaller than existing SOTA of 85k from TSPLib - a gap made exponentially larger by the combinatorial nature of the problem.

[Author Response · Irwan Bello · 25 Jan 2017]
**Summary of new changes to the paper**

We ask reviewers to have a look at the new version of the paper again given the changes outlined below:

- We state clearly in the abstract, introduction, and conclusion that our results are still far from the state-of-the-art (this includes adding an updated version of Figure 1 back into the introduction).

- We include the original KnapSack baselines back into the paper.

- We explain in details how the running time of the LKH baseline is obtained.

- We modify the statement on the performance of greedy approaches: instead of stating that they are “just a few percents from optimality”, we express that they are “still quite far from optimality”.

We thank reviewers for their help in improving the quality of the paper.

[Final Decision · Program Chairs · 06 Feb 2017]
**ICLR committee final decision**

This was one of the more controversial submissions to this area, and there was extensive discussion over the merits and contributions of the work. The paper also benefitted from ICLRs open review system as additional researchers chimed in on the paper and the authors resubmitted a draft. The authors did a great job responding and updating the work and responding to criticisms. In the end though, even after these consideration, none of the reviewers strongly supported the work and all of them expressed some reservations. 
 
 Pros:
 - All agree that the work is extremely clear, going as far as saying the work is "very well written" and "easy to understand". 
 - Generally there was a predisposition to support the work for its originality particularly due to its "methodological contributions", and even going so far as a saying it would generally be a natural accept.
 
 Cons:
 - There was a very uncommonly strong backlash to the claims made by the paper, particularly the first draft, but even upon revisions. One reviewer even saying this was an "excellent example of hype-generation far before having state-of-the-art results" and that it was "doing a disservice to our community since it builds up an expectation that the field cannot live up to" . This does not seem to be an isolated reviewer, but a general feeling across the reviews. Another faulting "the toy-ness of the evaluation metric" and the way the comparisons were carried out.
 - A related concern was a feeling that the body of work in operations research was not fully taken account in this work, noting "operations research literature is replete with a large number of benchmark problems that have become standard to compare solver quality". The authors did fix some of these issues, but not to the point that any reviewer stood up for the work.